# Changes of Phytochemical Components (Urushiols, Polyphenols, Gallotannins) and Antioxidant Capacity during *Fomitella fraxinea*–Mediated Fermentation of *Toxicodendron vernicifluum* Bark

**DOI:** 10.3390/molecules24040683

**Published:** 2019-02-14

**Authors:** Da-Ham Kim, Min-Ji Kim, Dae-Woon Kim, Gi-Yoon Kim, Jong-Kuk Kim, Yoseph Asmelash Gebru, Han-Seok Choi, Young-Hoi Kim, Myung-Kon Kim

**Affiliations:** 1Department of Food Science and Biotechnology, Chonbuk National University, Jeonju 54896, Jeonbuk, Korea; dadaham@naver.com (D.-H.K.); kmj6202@hanmail.net (M.-J.K.); kdwoon1@naver.com (D.-W.K.); seokmin0000@naver.com (G.-Y.K.); rlawodrnr@naver.com (J.-K.K.); yagebru@gmail.com (Y.A.G.); yhoi1307@hanmail.net (Y.-H.K.); 2Department of Agriculture and Fisheries Processing, Korea National College of Agriculture and Fisheries, Jeonju 54874, Jeonbuk, Korea; coldstone@korea.kr

**Keywords:** *Toxicodendron vernicifluum*, *Fomitella fraxinea*, fermentation, urushiols, polyphenols, antioxidant activity

## Abstract

The stem bark of *Toxicodendron vernicifluum* (TVSB) has been widely used as a traditional herbal medicine and food ingredients in Korea. However, its application has been restricted due to its potential to cause allergies. Moreover, there is limited data available on the qualitative and quantitative changes in the composition of its phytochemicals during fermentation. Although the *Formitella fraxinea*-mediated fermentation method has been reported as an effective detoxification tool, changes to its bioactive components and the antioxidant activity that takes place during its fermentation process have not yet been fully elucidated. This study aimed to investigate the dynamic changes of urushiols, bioactive compounds, and antioxidant properties during the fermentation of TVSB by mushroom *F. fraxinea*. The contents of urushiols, total polyphenols, and individual flavonoids (fisetin, fustin, sulfuretin, and butein) and 1,2,3,4,6-penta-*O*-galloyl-β-D-glucose (PGG) significantly decreased during the first 10 days of fermentation, with only a slight decrease thereafter until 22 days. Free radical scavenging activities using 2,2-diphenyl-1-picrylhydrazyl (DPPH), 2,2′-azino-bis(3-ethylbenzothiazoline-6- sulfonic acid) (ABTS), and ferric reducing/antioxidant power (FRAP) as an antioxidant function also decreased significantly during the first six to nine days of fermentation followed by a gentle decrease up until 22 days. These findings can be helpful in optimizing the *F. fraxinea*–mediated fermentation process of TVSB and developing functional foods with reduced allergy using fermented TVSB.

## 1. Introduction

*Toxicodendron vernicifluum* (TVSB, Stokes) F. Barkley (formerly known as *Rhus verniciflua* Stokes) belongs to the Anacardiaceae family [1], and is widely distributed in China, Japan, and Korea [2]. It is commonly known as the lacquer tree or vanish tree (also known as sumac). The stem and bark of *T. vernicifluum* have traditionally been used as folk medicines for treating blood disorders, hepatic disorders, gastric disorders, inflammatory diseases, anti-aging, paralysis, hypertension, and various cancers, as well as materials for lacquered chicken and duck soup recipes in Korea [3,4,5].

The xylem and bark of *T. vernicifluum* have also been reported to possess strong antioxidant [6,7], antiplatelet [8,9], immune-enhancing [10], neuroprotective [11,12], anti-inflammatory [12,13], and anti-cancer activities [4,14,15]. These health benefits were found to be attributed to the presence of flavonoids, gallotannins, and phenolic acids [3,16]. However, urushiol congeners found in the plant can cause allergic contact dermatitis with irritation, inflammation, and blistering in sensitive individuals [17,18]. Various methods have been attempted to remove or reduce urushiol congeners from the xylem and bark of *T. vernicifluum*, including physicochemical treatments such as solvent extraction, pyrolysis at high temperature, and enzymatic methods. Although some of these methods are effective at removing urushiols, they have limitations such as the generation of harmful substances, high cost, process complexity, and low efficiency [19].

A new biological method has been introduced to remove urushiols through fermentation with Basidiomycete *Formitella fraxinea* as an alternative method to solve these problems. This method was able to efficiently decrease more than 90% of the urushiols in TVSB [19]. Recently, an aqueous extract of urushiol-free fermented TVSB by *F. fraxinea* has been approved for use in soy sauce, fermented vinegar, and some alcoholic beverages by the Korea Food and Drug Administration (KFDA) [20], and its extracts are currently being marketed as functional health foods in Korea. Many studies on the biological activities of TVSB fermented by *F. fraxinea* have been reported, and most of these results were obtained by fermentation with *F. fraxinea* for 20–28 days [5,7,11,21,22]. While the fermentation method using *F. fraxinea* is being accepted as an effective tool to remove urushiols in TVSB, the dynamic changes of its bioactive components such as total polyphenols, individual phenolic compounds, gallotannins, and antioxidant activity during the fermentation process have not yet been fully elucidated. Flavonoids (fustin, fisetin, sulfuretin, and butein), PGG, and gallic acid as a gallotannin group are considered the major active constituents that are responsible for the various biological effects of TVSB [9,23,24,25]. Therefore, the objective of this study was to investigate dynamic changes in the polyphenols, individual phenolic compounds, gallotannins, and antioxidant activity during the fermentation of TVSB by mushroom *F. fraxinea.*

In this study, we demonstrated the optimization of an effective detoxification of urushiol congeners in TVSB through monitoring fermentation time while simultaneously intending to achieve a minimized loss of other useful phenolic compounds. The optimal fermentation period to achieve the desired detoxification effect was found to be 13–16 days. This finding provides valuable information to reconsider the common practice of fermenting TVSB extracts with *F. fraxinea* for 20–28 days prior to their applications in Korea.

## 2. Results and Discussion

### 2.1. Changes of Urushiols

Although TVSB has attracted much attention since ancient times in Korea due to its beneficial properties to human health, its allergenic urushiol congeners have limited its application in the food and pharmaceutical industries. Moreover, the contents of urushiols in *T. verniciflua* are higher in the bark than in its xylem. Therefore, it is necessary to remove its urushiol congeners during food or pharmaceutical applications. It has been reported that fermentation with *F. fraxinea* is an effective way to remove toxic urushiols from TVSB, and that mushroom laccases play an important role in the detoxification [19]. In this study, changes of urushiol congeners during *F. fraxinea*-mediated fermentation were analyzed by HPLC. As shown in Figure 1A, pentadecatrienylcatechol (C_15:3_) (671.08 ± 25.43 μg/g dry weight (DW)) was the most abundant, followed by pentadecenylcatechol (C_15:1_) (461.80 ± 4.95 μg/g DW) and pentadecadienylcatechol (C_15:2_) (95.77 ± 6.43 μg/g DW) in unfermented TVSB (UTVSB). The content of C_15:3_ rapidly decreased from 671.08 ± 25.43 μg/g DW in UTVSB to 115.22 ± 11.14 μg/g DW (82.8% decrease) and 48.61 ± 6.51 μg/g DW (92.8% decrease) after 10 and 13 days of fermentation, respectively. Thereafter, it showed only a slow decrease, reaching 30.66 ± 2.70 μg/g DW (95.4% decrease) at the end of the fermentation process (22 days). The C_15:2_ and C_15:1_ contents also decreased in a similar pattern to that of C_15:3_. The control was incubated for the same time intervals as the fermented sample. However, HPLC analysis of the controls was carried out only for 0 and 22-day time points, because the thin-layer chromatography (TLC) results (Appendix A) showed no difference among all of the matched controls. When the TVSB was treated under the sample conditions, but without *F. fraxinea* inoculation as the matched control, no meaningful changes were observed, even after 22 days compared to zero days (Appendix A). These results suggest that the gradual decreases of urushiols during fermentation are due to the action of related enzymes secreted from *F. fraxinea*.

The total content of urushiol congeners decreased from 1228.7 ± 34.6 μg/g DW in UTVSB to 186.02 ± 24.3 μg/g DW (84.9%), 78.77 ± 11.2 μg/g DW (93.6%), and 47.75 ± 9.5 μg/g DW (96.1% decrease) at the end of 10, 13, and 22 days of fermentations, respectively (Figure 1B). Considering these results, the optimal fermentation period for the removal or reduction of urushiol congeners from TVSB by *F. fraxinea* is suggested to be between 13–16 days of fermentation. Choi et al. [19], who attempted a similar biological method for the reduction of urushiols, reported that the content of C_15:3_ decreased rapidly in the first five days of fermentation, followed by a gradual decrease, reaching the lowest levels after 15 days of fermentation.

The degradation of urushiols is closely associated with the laccase that is secreted from *F. fraxinea* mycelia. Laccase is a type of copper-containing polyphenol oxidase (*p*-diphenol oxidase or benzenediol; oxygen oxidoreductase, EC 1.10.3.2). More than 100 fungal laccases have been purified and characterized from *Basidiomycetes* and *Ascomycetes* [26]. Since these enzymes can catalyze the oxidation of phenolic compounds such as polyphenols, methoxy-substituted phenols, *o*-diphenols, *p*-diphenols, aromatic amines, and syringaldazine, it is somewhat difficult to categorize laccases based on the type of substrates, because a wide range of substrates can be catabolized [26,27,28]. However, laccase can oxidize urushiols and result in the formation of semiquinone radicals under aerobic conditions. This is a typically unstable product, and it undergoes subsequent changes either through an attack on the urushiol nucleus, forming biphenyl compounds, or a disproportionation reaction to give urushiol quinone [29]. It may also undergo polymerization reactions, which results in the formation of a conjugated insoluble product. All of these mechanisms lead to the reduction of the active site of urushiols, thereby resulting in its detoxification [29,30].

### 2.2. Change of Total Phenol, Total Flavonoid, and Individual Flavonoid

The diverse pharmacological activities of *T. vernicifluum* are believed to be mainly attributed to the presence of phenolic compounds such as fustin, fisetin, sulfuretin, quercetin, taxifolin, garbanzol, butein, gallic acid, dihydroxybenzoic acids, and other phenolic compounds [24,25]. It is known that fustin, fisetin, sulfuretin, and butein are abundant in the bark and xylem of *T*. *vernicifluum* [12,13]. Therefore, it is necessary to minimize the degradation of these valuable bioactive components during fermentation. Changes to the total phenol and total flavonoid contents in TVSB during *F. fraxinea*-mediated fermentation were investigated.

The total phenol content decreased sharply from 2927.6 ± 228.8 mg GAE/100 g DW in UTVSB to 859.2 ± 156.6 mg GAE/100 g DW after 10 days of fermentation, reaching 599.3 ± 78.1 mg GAE/100 g DW at the end of the fermentation (Figure 2A). The total flavonoid content decreased from 1365.4 ± 130.6 mg RE/100 g DW in UTVSB to 598.3 ± 64.5 mg RE/100 g DW and 272.8 ± 39.6 mg RE/100 g DW after 10 and 22 days of fermentation, respectively (Figure 2B). The reduction of total phenol and total flavonoid during fermentation is closely associated with the polyphenol oxidases that are secreted from *F. fraxinea* mycelia. Polyphenol oxidases (EC 1.14.18.1) that are known as tyrosinase, laccase, catechol oxidase, catecholase, phenolase, cresolase, and urushiol oxidase based on substrate specificity are known to have considerable overlap in substrate affinities [31,32]. These enzymes are distributed in a wide range of fungi, higher plants, and mammals. They can catalyze the oxidation of a broad range of phenolic compounds, including phenols, phenolic acids, flavonoids, tyrosine, l-3,4-dihydroxyphenylalanine (l-DOPA), naphthols, bisphenols, and other phenolic compounds [28,31,33].

In this study, changes of individual flavonoids during fermentation were also monitored by HPLC analysis (Figure 3A). The contents of major flavonoids such as fustin, fisetin, sulfuretin, and butein showed noticeable decreases at the initial stage (five days) of fermentation, followed by gradual decreases up until 22 days (Figure 3B). It has been previously reported that basidiomycete polyphenol oxidases can effectively utilize flavonoids such as quercetin, fisetin, rutin, luteolin, myricetin, and kaempferol, as well as phenolic acids as substrates [28,31,33,34]. Therefore, the reduction of flavonoids that was observed during the fermentation of the TVSB extract is suggested to be mediated by polyphenol oxidases that are secreted from *F. fraxinea*.

### 2.3. Changes of PGG

Gallotannin appears to have important biological roles including antioxidant, antibacterial, anti-inflammatory, anti-hypoglycemic, anti-angiogenic, and anti-cancer activities. 1,2,3,4,6-penta-O-galloyl-β-D-glucose (PGG), which is a gallotannin, are widely distributed in fruit, berries, and woody plants. PGG and its hydrolysates are among the main groups of antioxidant polyphenols in UTVSB. They have attracted attention in recent years due to their beneficial properties for human health [16,23]. In this study, PGG, gallic acid, and methyl gallate were detected as gallotannins in UTVSB.

PGG content decreased from 5590.0 ± 324.5 μg/g DW in UTVSB to 1580.0 ± 282.2 μg/g DW (71.7% decrease) and 1452.9 ± 211.9 μg/g DW (74.0% decrease) after 10 and 13 days of fermentation, respectively. Only slight changes in PGG content were observed from day 13 to day 22 of fermentation (Figure 4). Gallic acid and methyl gallate contents also showed noticeable decreases within seven days from the start of fermentation.

### 2.4. Characterization of PGG Hydrolysates by HPLC-MS

Gallotannins are important bioactive compounds in medicinal plants. PGG is especially abundant in the TVSB [25]. This compound showed a consistent decrease during the initial stage of fermentation, as shown in Figure 4. HPLC analysis (Figure 5B) revealed the formation of new peaks in the 80% methanol extract of fermented TVSB that were not detected in the similar extract of UTVSB (Figure 5A). These peaks were then tentatively identified by HPLC-MS analysis. Total ion current (TIC) chromatograms of the 80% methanol extract from 10-day fermented TVSB and the hydrolysate of PGG by crude enzyme preparation isolated from fermented TVSB are shown in Figure 6. Forty-two compounds in the fermented TVSB and 40 compounds in PGG hydrolysate were identified tentatively (Table 1).

The peak **1** at 0.714 min was identified as gallic acid based on a quasimolecular ion at *m/z* 169.0 [M − H]^−^. Peaks **2** and **4**, which were detected in both samples, showed the same quasimolecular ions at *m/z* 331.1 [M − H]^−^ in the negative electrospray ionization mass spectrometry (ESI-MS) spectra and λ_max_ at 276 nm. These compounds possessed one hexosyl moiety (162 amu) more than the fragment of peak **1** (gallic acid). Thus, these compounds were characterized as monogalloylglucose (MGG) isomers. Thirteen isomers (peaks **3, 5, 6, 11, 12, 13, 16, 17,** and **19**–**23**) detected in fermented TVSB showed a quasimolecular ion at 483.1 [M − H]^−^ with λ_max_ at 273–276 nm.

These compounds possessed one galloyl moiety (152 amu) more than monogalloylglucose. Accordingly, these compounds were tentatively characterized as digalloylglucose (DGG) isomers. In a similar manner, nine isomers (peaks **24, 26**–**31**, **33**, and **35**) showed quasimolecular ions at *m/z* 635.1 [M − H]^−^ and similar UV profiles. They possessed one galloyl moiety (152 amu) more than DGG. These compounds were characterized as trigalloylglucose (TGG) isomers. Six isomers at peaks **32**, **34**, and **37**–**40** were also suggested as tetragalloylglucose (TeGG) isomers by their quasimolecular ion at *m/z* 787.1 [M − H]^−^ and UV profiles (λ_max_ at 273–276 nm). The mass spectra of oligomeric galloylglucoses that were identified in both samples are presented in Figure 7.

Some of these PGG hydrolysates have been previously found in *Euvcalyptus nitens* wood [35], mango [36], *Rhus coriaria* [37], *Psidium guineense, Syzygium cumini, Pouteria macrophylla* [38], and the degradation products of tannin by *Aspergillus niger* [39]. These compounds can affect the expression of biological activities of target plants. In addition, galloylglucose oligomers may have more potent antioxidant activities than PGG or ascorbic acid [40]. Generally, microbial tannase (E.C.3.1.1.20) hydrolyses the ester and depside bonds of gallotannins and PGG to produce gallic acid and methyl gallate as end products [41,42]. However, the PGG-degrading enzyme from *F. fraxinea* in this study catalyzed the hydrolysis of the bonds that were present in the molecules of PGG to produce oligomeric galloylglucoses—mainly DGG, TGG, and TeGG—instead of gallic acid. In addition, peak **36** was tentatively identified as ellagic acid based on its quasimolecular ion at 301.0 [M − H]^−^ and its UV profile. Peaks **42**, **46**, and **48** with quasimolecular ions [M − H]^−^ of *m/z* 285.1, *m/z* 301.1, and *m/z* 271.1 were identified as fisetin, quercetin, and butein, respectively. These compounds have been previously reported as the major bioactive components of *T. vernicifluum* [24,25].

### 2.5. Changes of Antioxidant Activities

Due to the differences in the theoretical bases of different antioxidant measurements, a single antioxidant property model can hardly reflect the antioxidant capacity of the samples [43]. For this reason, three model systems, i.e., 2,2-diphenyl-1-picrylhydrazyl (DPPH) radical scavenging activity, 2,2′-azino-bis(3-ethylbenzothiazoline-6- sulfonic acid) (ABTS) radical scavenging activity, and ferric reducing/antioxidant power (FRAP) were used to evaluate the antioxidant properties of fermented TVSB. DPPH and ABTS rely on the reaction of radicals and cation radicals, respectively, and the FRAP method relies on the reduction by the antioxidant components of complex ferric ion–TPTZ (2,4,6-tri(2-pyridyl)-*s*-triazine). A study by Gorinstein et al. [44] observed a high correlation of antioxidant capacities measured by ABTS, DPPH, and FRAP assays in some fruits, while Pellegrini et al. [45] reported a weak correlation on antioxidant capacity measured by FRAP and ABTS assays in some vegetables and beverages. The results of changes in antioxidant capacities are presented in Figure 8. DPPH radical scavenging activity rapidly decreased from 82.45 ± 1.41% in unfermented TVSB (day 0) to 21.31 ± 1.35% after nine days and 15.88 ± 0.85% after 12 days of fermentation, while no significant change was observed thereafter until 22 days. The antioxidant properties of *T. vernicifluum* stem and bark can be explained by the higher contents of phenolic compounds such as fustin, fisetin, sulfuretin, quercetin, gallotannins, and urushiols [6,7].

ABTS radical scavenging activity as an inhibition percentage (%) showed a significant decrease (*p* < 0.05) from 99.83 ± 1.96% in the unfermented sample to 67.85 ± 9.40% and 53.04 ± 2.11% after nine and 15 days of fermentation, respectively. No meaningful change was observed thereafter until 22 days. Trolox equivalent antioxidant capacity (TEAC) is a method that provides information on the overall status of antioxidants within a test sample, and has proven to be a useful indicator for determining the ability of an organism to mitigate the potential damage caused by reactive oxygen species. It has been widely used for studying the antioxidant capacity of phytochemicals and biological samples [46]. In this study, the TEAC value also decreased from 9.89 ± 0.06 mM TE/g DW in the unfermented sample to 6.82 ± 0.03 mM TE/g DW (31% decrease) and 6.29 ± 0.03 mM TE/g DW (36.4% decrease) after nine and 15 days of fermentation, respectively, with no significant change until 22 days. Similarly, the FRAP value decreased significantly (*p* < 0.05) during the first 12 days of fermentation. It decreased by 57.3% (from 18.39 ± 0.10 to 7.85 ± 0.21 mM/g DW), with no significant decrease observed thereafter until 22 days.

Phenolic compounds can be enzymatically oxidized and polymerized by polyphenol oxidases such as laccases or tyrosinases. Previous studies have reported that oxidized products, polymers, and aggregates exhibit stronger antioxidant activities than their corresponding monomeric flavonoids [34,47,48,49]. However, it can be clearly observed that fermented TVSB still retained antioxidant activities, although it was relatively weaker compared to that of UTVSB. Kim et al. [7] reported that the methanol and ethyl acetate fractions of fermented TVSB by *F. fraxinea* showed higher radical scavenging and reducing power activities than those of synthetic antioxidant butylated hydroxyanisole (BHA) and butylated hydroxytoluene (BHT). Therefore, fermented TVSB might be more preferable than the above synthetic antioxidant products. Oligomeric galloylglucoses that originated from PGG such as MGG, DGG, TGG, and TeGG may also have contributed to the antioxidant activities of fermented TVSB.

## 3. Materials and Methods

### 3.1. Plant Material

TVSB was collected from a 10-year-old tree that had been cultivated in Imsil-Gun, Chonbuk Province, Republic of Korea in October 2015. The plant material was authenticated by one of the authors (M.K. Kim). A voucher was deposited at the Fermentation Laboratory, Department of Food Science and Biotechnology, Chonbuk National University, Jeonju 54896, Jeonbuk, Republic of Korea. The fresh TVSB was air-dried (60 °C) for 24 hours and kept at room temperature (20 °C) until used for further experiments within one week.

### 3.2. Chemicals

Urushiol standards (C_15:3_, C_15:2_, C_15:1_) were purchased from Phytolab GmbH and Co. (Dutendorfer Straße, Vestenbergsgreuth, Germany). Fustin, fisetin, sulfuretin, and butein were purchased from Chromadex Co. (Irvine, CA, USA). Gallic acid, methyl gallate, quercetin, taxifolin, PGG, Folin-Ciocalteu reagent, 2,2-diphenyl-1-picrylhydrazyl (DPPH), 2,2′-azino -bis(3-ethylbenzothiazoline-6-sulfonic acid), diammonium salt (ABTS), 6-hydroxy-2,5,7,8 -tetramethylchromane-2-carboxylic acid (Trolox), 2,4,6-tri(2-pyridyl)-*s*-triazine (TPTZ), potassium persulfate, tannic acid, bovine serum albumin (BSA), and sodium dodecyl sulfate (SDS) were purchased from Sigma-Aldrich (St. Louis, MO, USA). HPLC grade methanol, acetonitrile, and deionized water were purchased from J.T. Baker Co. (Phillipsburg, NJ, USA). All of the other reagents were of analytical grade.

### 3.3. Microorganism and Culture Conditions

The strain of *Fomitella fraxinea* (Bull.) Imazeki (KACC 42289) was kindly donated by the Korean Agricultural Culture Collection (KACC) of the Rural Development Administration (RDA), Wanju, Jeonbuk, Republic of Korea.

### 3.4. Fermentation of TVSB by F. fraxinea

The strain was preincubated onto potato dextrose agar (Becton, Dickinson and Company, Sparks, MD, USA) for six days at 25 °C. Sterilization of the culture media was performed at 121 °C for 30 min. The preincubated strain was inoculated into germinated-malt medium (11 Brix°) saccharified at 65 °C with four-fold tap water (*v/v*) for eight hours, and then cultured for two weeks at 25–26 °C with gentle shaking (120 rpm) using an orbital shaker (model SK-600, Jeiotech Co., LTD. Daejon, Korea). The air-dried TVSB of crushed coarse powder was placed in an Erlenmeyer flask (500 mL). The moisture content of each powdered sample (100 g) was then adjusted to approximately 65% with tap water. Each flask was sterilized at 121 °C for 30 min. The day-0 (control) sample was freeze-dried immediately and stored at −20 °C until further analysis. The remaining samples were inoculated with five mL of *F. fraxinea* liquid culture except for the unfermented (matched controls) and incubated at room temperature (25–26 °C) with gentle shaking (120 rpm) for each required time frame before further analysis. All of the fermentations were set up on the same day, and each sample for the specific time point was retrieved with its matched control at the same time. Each sample was then freeze-dried at the end of each time frame and stored until use.

### 3.5. Extraction and HPLC Analysis of Urushiols.

#### 3.5.1. Extraction

A powdered sample (1.0 g) of UTVSB and fermented TVSB was extracted twice with 20 mL of acetone in an ultrasonic bath (Hwa Shin Instrument Co., Ltd., Seoul, Korea) for 20 min at room temperature and centrifuged at 5000 rpm for 15 min. Supernatants were combined and concentrated under vacuum at 45 °C, and the residue was dissolved in acetone (five mL).

#### 3.5.2. Thin-Layer Chromatography (TLC) Analysis

TLC was carried out for the extracted samples as follows. The developing solvent was a mixture of chloroform–methanol–water (65:35:10, lower phase). The spots were detected either under UV (254 nm) or by spraying a 10% CuSO_4_ solution in an 8% H_2_SO_4_ solution followed by heating at 110 °C for 10 min.

#### 3.5.3. HPLC Analysis

The standard stock solution of urushiols was prepared at a concentration of 1000 μg/mL in acetone. The stock solution was serially diluted with acetone to obtain a calibration curve at seven concentration levels (10–500 μg/mL) for C_15_ and C_15:1_ and at 5–250 μg/mL for C_15:2_. HPLC analysis was performed using an HPLC system (Waters, Milford, MA, USA) equipped with a 2690 separation module and 996 photodiode array (PDA) detector with a YMC-Pak Pro C18 column (4.6 mm × 250 mm, five μm; YMC Co., LTD, Tokyo, Japan). The mobile phase was 90% MeOH in deionized water at a flow rate of 1.0 mL/min (isocratic) with an injection volume of 10 μL. The UV detection wavelength was set at 273 nm. The concentration of each constituent was calculated using the calibration curve by plotting the peak area of the corresponding substance against the concentration (in μg/mL) of the standard substance.

### 3.6. Extraction for Polyphenols and Gallotannins

A powdered sample (1.0 g) of UTVSB and fermented TVSB was extracted twice with 20 mL of 80% aqueous MeOH using an ultrasonic bath for 30 min and centrifuged at 5000 rpm for 15 min. The supernatants was concentrated at 45 °C under reduced pressure, and the residue was dissolved in 80% methanol (five mL) for an analysis of total flavonoids, individual flavonoids, and gallotannins by HPLC, and for antioxidant activity assay.

### 3.7. Total Phenol

The total phenolic content of the samples was determined according to a method described by Chandra et al. [50] with some modifications. Briefly, an 80% MeOH extract (20 μL) of each sample was mixed with a 50% Folin–Ciocalteu phenol reagent (20 μL) in 96-well plates. After five minutes, one N of sodium carbonate (20 μL) was added to the mixture, and distilled water was added to reach a final volume of 200 μL. After incubation at room temperature in the dark for 30 min, the absorbance of a test sample against a blank was measured at a wavelength of 725 nm using a VersaMax ELISA microplate leader (Molecular Devices, LLC, CA, USA). The phenolic content was calculated based on a calibration curve of gallic acid. The result was expressed as mg of gallic acid equivalent (GAE) per 100 g of the dried sample.

### 3.8. Total Flavonoid

The total flavonoid content was measured according to a method described by Zhishen et al. [51] with some modifications. Briefly, an 80% MeOH extract (30 μL) of each sample was mixed with 30 μL of 5% sodium nitrite solution. After five minutes of reaction, 300 μL of 5% aluminum chloride was added. Then, 200 μL of one N of NaOH was added six minutes later, and the total volume was adjusted to one mL with distilled water. The absorbance of the test sample against a blank was measured at a wavelength of 510 nm with a Shimadzu UV-1601 spectrophotometer (Kyoto, Japan). The flavonoid content was calculated using a calibration curve of rutin. The result was expressed as mg rutin equivalent (RE) per 100 g of the dried sample.

### 3.9. HPLC Analysis of Individual Flavonoids

The stock solution was serially diluted with methanol to obtain seven concentration levels (12.5–750 μg/mL) for each analyte to prepare a calibration curve. HPLC analysis was performed using an HPLC system (Waters, Milford, MA, USA) equipped with a 2690 separation module and a 996 photodiode array (PDA) detector with a ZORBAX Eclipse XDB C18 column (4.6 mm × 250 mm, five μm; Agilent Technologies, Technologies, Inc., Santa Clara, CA, USA). The mobile phase was 0.1% formic acid in ionized water (A) and 90% acetonitrile in water (B). The ratio of the mobile phase was A:B mixed at 90:10 (zero to one minute), 20:80 (one to 15 min), and 90:10 (15–25 min) at a flow rate of 1.0 mL/min. The UV detection wavelength was set at 280 nm. The concentration of each constituent was calculated using a calibration curve by plotting the peak area of the corresponding substance against the concentration (μg/mL) of the standard substance.

### 3.10. Hydrolysis of PGG by Tannases

#### 3.10.1. Preparation of PGG

PGG was prepared from tannic acid according to the method of Chen and Hagerman [52] with slight modification. A sample of 5.0 g of tannic acid was methanolyzed in 100 mL of 70% aqueous methanol in 0.1 M of sodium acetate (pH 5.00 at 65 °C) After methanolysis for 15 h, the pH of the reaction mixture was adjusted to 6.0, and methanol was removed by evaporation under reduced pressure below 35 °C. Water was added to maintain the volume, replacing evaporated methanol. The resulting aqueous solution was sequentially extracted with three volumes of diethyl ether and ethyl acetate. The ethyl acetate fraction was combined and evaporated under reduced pressure, and then, distilled water was added to the ethyl acetate extract. The resulting milky suspension was centrifuged for 15 min at 4500 rpm (model VS-550, Vision Scientific Co., LTD, Daejon, Korea). The precipitate containing PGG was washed twice with 20 mL of ice-cold 2% aqueous methanol. In the last step, the final material was obtained by freeze-drying. The material obtained was identified as PGG by LC-MS and NMR spectroscopy (JEOL JNM-ECA 600 FT-NMR, Akishima, Tokyo, Japan) as follows: UV, λ_max_ 280 nm (methanol); ESI-LC-MS, *m/z* 939.2 [M − H]^−^ (C_41_H_32_O_26_); ^1^H−NMR (600 MHz, methanol-*d*_4_), δ 7.01 (2H, s), 6.95 (2H, s), 6.87 (2H, s), 6.84 (2H, s), 6.79 (2H, s), 6.13 (1H, d, *J* = 8.25 Hz), 5.81 (1H, t, *J* = 9.62 Hz), 5.51 (1H. dd, *J* = 9.62 Hz), 5.50 (1H, t, *J* = 9.62 Hz), 4.42 (1H, d, *J* = 11.68 Hz), 4.30 (2H, m); ^13^C−NMR (150 Hz, methanol-*d*_4_), δ 168.05, 167.41, 167.14, 167.04, 166.33, 146.66 (x2), 146.58, 146.54, 146.48, 146.39, 140.87, 140.47 (x2) 140.41(x2), 140.24 (x2), 140.12 (x2), 121.15, 120.47, 120.34, 120.31, 119.84, 110.73 (x2), 110.57 (x2), 110.51 (x2), 110.48 (x2), 110.44 (x2), 93.93, 74.53, 74.22, 72.30, 69.91, 63.22.

#### 3.10.2. Preparation of Crude Enzymes from Fermented TVSB

All of the procedures for the preparation of crude enzyme were carried out at 4 °C unless otherwise indicated. TVSB (50 g) that had been fermented for 10 days with *F. fraxinea* was mixed with 500 mL of 0.1 M of sodium phosphate buffer (pH 5.5) with gentle stirring for four hours, and then homogenized with an Omni mixer homogenizer (Omni International, Kennesaw, GA, USA) for one minute. The homogenate was squeezed through a cheese cloth, and the filtrate was centrifuged at 10000 × *g* for 20 min. The crude enzyme containing tannase was prepared by solid ammonium sulfate (30–80%) saturation. After centrifugation at 10000 × *g* for 20 min, the precipitate was dissolved in 10 mM of sodium acetate buffer (pH 5.5). After dialysis for 24 hours, the solution was centrifuged at 10000 × *g* for 20 min, and the supernatant was lyophilized.

#### 3.10.3. Tannase Assay

Lyophilized crude enzyme preparation (50 mg) was dissolved in 10 mL of 0.1 M of sodium acetate buffer (pH 5.5). Tannase activity was measured using the method of Mondal et al. [53] with modifications. Briefly, the solution (150 μL) containing three mM of tannic acid (0.2 M of acetate buffer, pH 5.5) as a substrate was added to 200 μL of crude enzyme, and the reaction solution was incubated at 40 °C for 30 min. Subsequently, a BSA solution (1 mg/mL) was added to the flask and then kept at room temperature for 15 min. The reaction solution was centrifuged (5000 rpm, 20 min) to remove the supernatant, and 1.5 mL of SDS–triethanolamine solution was added to the precipitate. To stabilize the color, 0.5 mL of 0.16% FeCl_2_ (0.01 N HCl) was added, and the mixture was allowed to stand at room temperature for 15 min. Its absorbance was then measured at a wavelength of 530 nm. The calibration curve was prepared using tannic acid under the same conditions. One unit (U) of tannase was defined as the amount of enzyme that was required to hydrolyze 1.0 μM of tannic acid per minute under specified conditions.

#### 3.10.4. Enzymatic Hydrolysis of PGG

The reaction mixture (50 mL) containing 0.5 g of PGG in 2.5 mL of methanol and crude enzyme preparation (30–80% ammonium sulfate precipitate) isolated from fermented TVSB or *Asp. oryzae* tannase containing 30 U of tannase activity in 47.5 mL of 0.1 M sodium acetate buffer (pH 5.5) were incubated for 12 hours at 45 °C with gentle shaking. After the reaction mixture was kept in a boiling water bath for 10 min, it was subjected to HPLC and LC-MS analysis.

### 3.11. Analysis of PGG Hydrolysates in Fermented TVSB

#### 3.11.1. HPLC

The standard solution was serially diluted with 80% methanol to obtain seven concentration levels (12.5–750 μg/mL) for each compound to prepare a calibration curve. HPLC analysis was performed using an HPLC system equipped with a 2690 separation module and a 996 photodiode array (PDA) detector with a ZORBAX Eclipse XDB−C_18_ column. The mobile phase consisted of 0.1% formic acid in ionized water (A) and 90 % acetonitrile in water (B). The ratio of A:B as the mobile phase was maintained at 95:5 (zero to two minutes), 45:55 (two to 25 min), 40:60 (25 to 30 min), and 95:5 (30 to 40 min) at a flow rate of 0.8 mL/min. The UV detection wavelength was set at 310 nm.

#### 3.11.2. HPLC-MS

The powder of fermented TVSB (1.0 g) was extracted with 20 mL of acetone in an ultrasonic bath (Hwa Shin Instrument Co., Ltd., Seoul, Korea) for 20 min at room temperature and centrifuged at 4500 rpm for 20 min. The residue was extracted one more time using the same solvent. The supernatant was concentrated under vacuum at 45 °C, and the residue was dissolved in five mL of acetone. A standard stock solution of urushiols was prepared at a concentration of 1000 μg/mL in acetone. The stock solution was serially diluted with acetone to obtain a calibration curve at seven concentration levels (10–500 μg/mL) for C_15_ and C_15:1_ , and at 5–250 μg/mL for C_15:2_.

HPLC-MS analysis was performed on an Agilent chromatographic system 1100 series (Agilent Technologies, Santa Clara, CA, USA) equipped with a G1379A degasser, a G1312A binary pump, a 1329A autosampler, and a G1316A column thermostat. The LC was coupled with an Agilent ion trap 1100 SL mass detector. Separation was performed using a reverse phase column Kinetex C_18_ (50 × 2.1 mm i. d., 2.6 μm, Phenomenex, Torrance, CA, USA). The mobile phase consisted of 0.1% formic acid in water (A) and 0.1% formic acid in 90% acetonitrile with the following gradient program: zero to four minutes, 90% A; four to 25 min, 90–40% A; 25 to 30 min, 40–90% A; and 30 to 35 min, 90% A. The flow rate was set at 0.5 mL/min, and the temperature was set at 40 °C. The injection volume was two μL. The MS was equipped with an electrospray ionization (ESI) interface in negative ion mode. The ESI parameters were set as follows: ion source temperature, 350 °C; gas flow rate (nitrogen), 15 L/min; nebulizer pressure, 40 psi pressure; capillary voltage, 3000 V; fragmentation voltage, 400 V; collision energy, 0 V; and full-scan data acquisition, 50–1200 *m/z*.

### 3.12. Antioxidant Activity

#### 3.12.1. DPPH Free Radical Scavenging Activity

The DPPH radical scavenging activity of the sample was determined according to the method described by Thaipong et al. [54] with some modifications. Briefly, the extract (20 μL) was added to an 80-μL DPPH solution (500 μM) and 100-μL Tris-HCl buffer (0.1 M). The mixture was incubated at room temperature in the dark for 20 min. As a blank, the test was repeated using a buffer instead of a sample. Absorbance was measured at a wavelength of 517 nm using a microplate leader, and the scavenging activity of the extract was calculated against a blank as follows.
DPPH radical scavenging activity (%) = (1 − A_0_/A_1_) × 100(1)
where A_0_ and A_1_ are the absorbance values of the test sample and control, respectively.

#### 3.12.2. ABTS Free Radical Scavenging Activity

ABTS free radical scavenging activity was determined by the methods described by Thaipong et al. [54] with some modifications. Briefly, a mixture of ABTS (7.4 mM) solution and potassium persulfate (2.6 mM) solution in equal volumes was kept for 12 hours at room temperature in a dark to form an ABTS cation. The solution was diluted by mixing with methanol to obtain an absorbance of 1.0 ± 0.02 at 734 nm using a UV-Vis spectrophotometer. Then, 100 μL of the extract was added to 1400 μL of the diluted ABTS solution, and the mixture was incubated at room temperature for one hour in the dark. After the reaction, its absorbance was measured at a wavelength of 734 nm. The calibration curve was linear between 25–400 μM Trolox. Results were expressed as μM Trolox equivalent (TE) g dry weight, and the ABTS radical scavenging activity (%) was also calculated with the following equation:ABTS radical scavenging activity (%) = (1 − A_0_/A_1_) × 100(2)
where A_0_ and A_1_ are the absorbance values of the test sample and control, respectively.

#### 3.12.3. Ferric Reducing/Antioxidant Power (FRAP)

Ferric reducing power was determined using FRAP assay [55] with some modification. The working solution for FRAP assay was prepared by mixing 10 volumes of 300 mM of acetate buffer, pH 3.6, with one volume of 10 mM of TPTZ in 40 mM of HCl, and with one volume of 20 mM of ferric chloride. All of the required solutions were freshly prepared before their uses. A sample extract (80 μL) was added to 1420 μL of FRAP reagent. The reaction mixture was incubated at room temperature for 30 min in the dark. Then, the absorbance of the samples was measured at 593 nm. The calibration curve was linear between 25–600 μM Trolox. Results were expressed as μM Trolox equivalent (TE)/g dry weight.

### 3.13. Statistical Analysis

All of the experiments were performed in triplicate. All of the values were expressed as mean ± standard deviation (SD). All of the statistical analyses were performed with SPSS (ver. 10.1) for Windows. One-way analysis of variance (ANOVA) and Duncan’s multiple range test was carried out to test any significant difference among various treatments. Significant differences were determined at *p* < 0.05.

## 4. Conclusions

Dynamic changes in urushiol congeners, polyphenols, gallotannins, and antioxidant activities during the fermentation of TVSB by mushroom *F. fraxinea* to reduce or detoxify urushiols were investigated. The content of urushiol congeners noticeably decreased within 10 days of fermentation with a slow decrease thereafter until 22 days. The contents of total phenol, total flavonoid, and individual flavonoids also decreased in similar patterns as those of the urushiols. PGG was hydrolyzed during the fermentation process, resulting in the formation of a number of oligomeric galloylglucoses isomers. These results indicate that PGG was mainly hydrolyzed into oligomeric galloylglucoses (TeGG, TGG, DGG, and MGG) by tannin-hydrolyzing enzymes secreted from *F. fraxinea* mycelia during fermentation. The overall decrease in antioxidant activity during the fermentation of TVSB may be associated with a decrease in phenolic compounds. Nonetheless, the fermented TVSB has to some extent retained strong antioxidant activity, although it is relatively weaker than that of UTVSB. Oligomeric galloylglucoses such as MGG, DGG, TGG, and TeGG, which originated from PGG may also have contributed to the antioxidant activity of fermented TVSB. Although TVSB extracts are fermented for 20–28 days using *F. fraxinea* to be marketed as functional food raw materials in Korea, the present study suggests that the optimal fermentation period to remove or reduce toxic urushiols is between 13–16 days, with a minimized reduction of useful constituents such as flavonoids and gallotannins. The findings in the present work also provide clear evidence for the need of additional studies on the enzymatic characteristics of PGG-hydrolyzing enzymes secreted from *F. fraxinea* during the fermentation of TVSB.

## Figures and Tables

**Figure 1 molecules-24-00683-f001:**
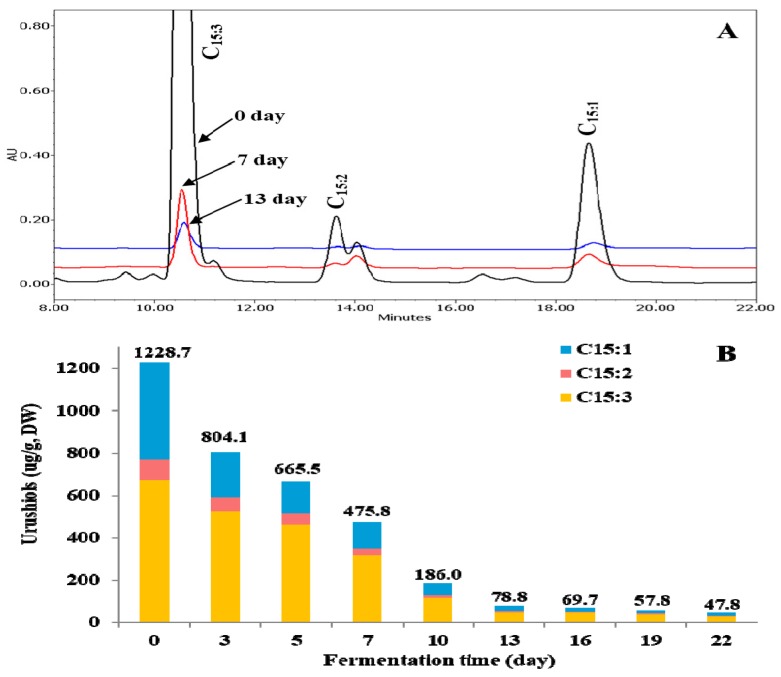
Changes of urushiol congeners during fermentation of *Toxicodendron vernicifluum* stem bark (TVSB) by *F. fraxinea.* Representative HPLC chromatograms (**A**) and contents (**B**).

**Figure 2 molecules-24-00683-f002:**
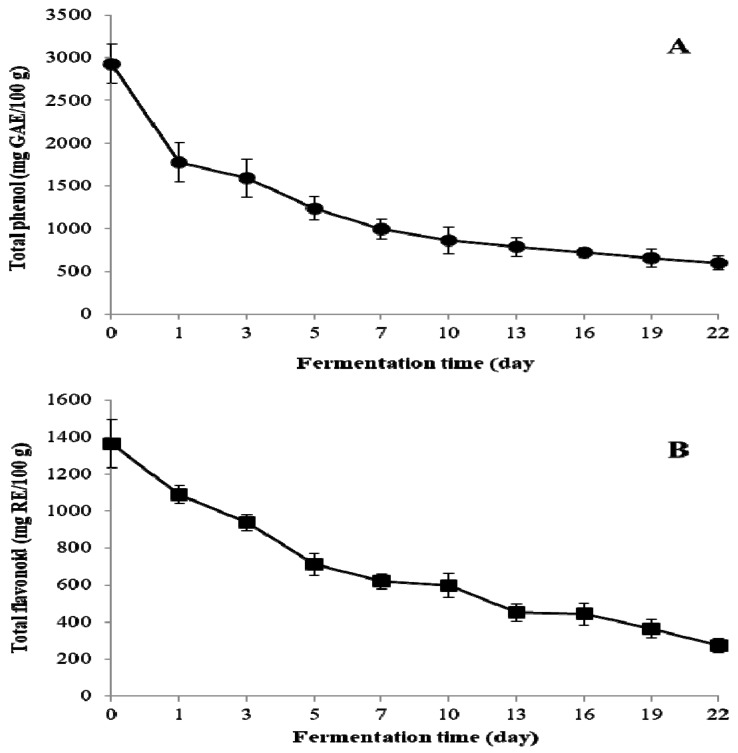
Changes in the contents of total phenol (**A**) and total flavonoid (**B**) during the fermentation of TVSB by *F. fraxinea*. Error bars are the standard deviations of triplicate measurements.

**Figure 3 molecules-24-00683-f003:**
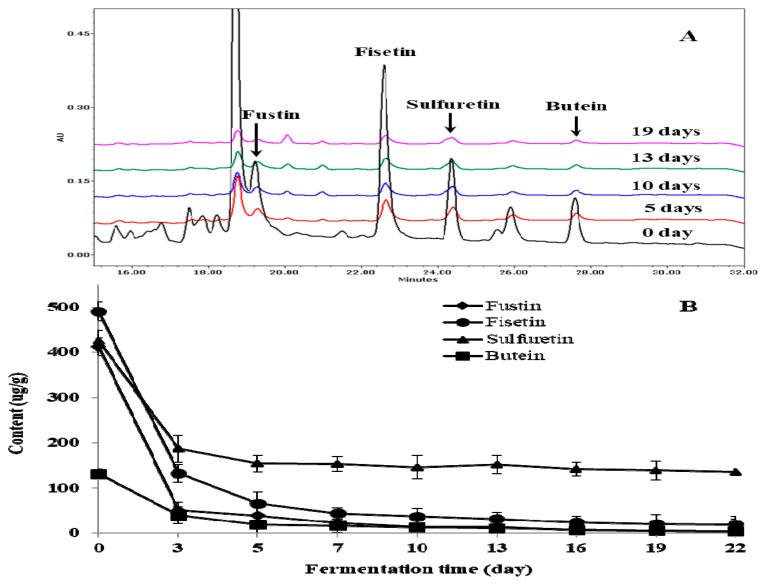
Changes of individual flavonoid compounds during the fermentation of TVSB by *F. fraxinea.* HPLC chromatograms (**A**), contents of individual flavonoids (**B**). Error bars are the standard deviations of triplicate measurements.

**Figure 4 molecules-24-00683-f004:**
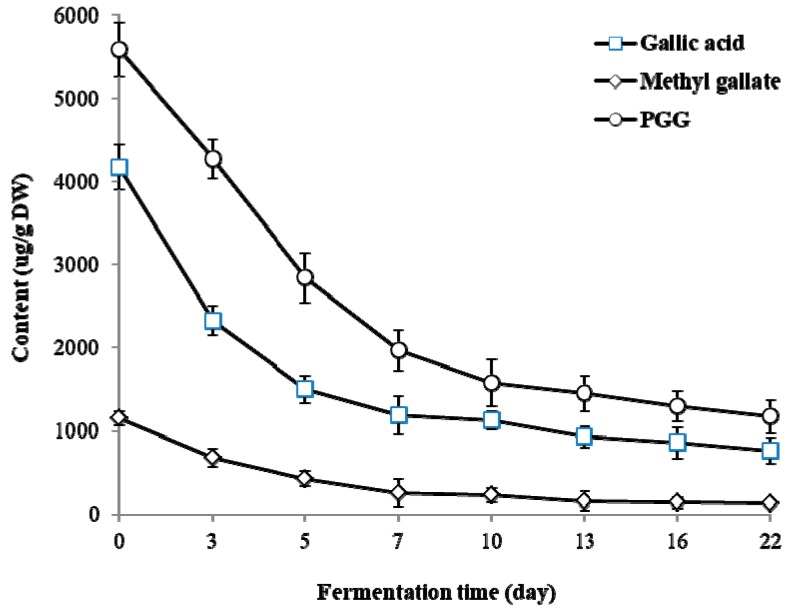
Changes of 1,2,3,4,6-penta-O-galloyl-β-D-glucose (PGG) and its metabolites gallic acid and methyl gallate during the fermentation of TVSB by *F. fraxinea*. Error bars are the standard deviations of triplicate measurements.

**Figure 5 molecules-24-00683-f005:**
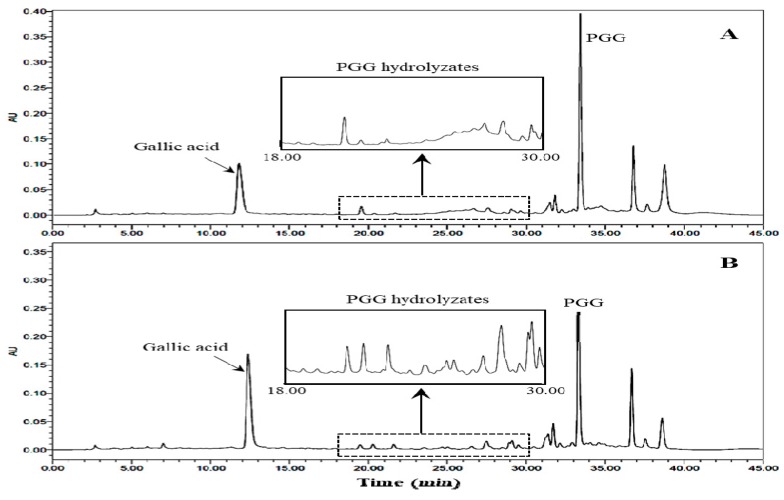
HPLC profiles (at 310 nm) of 80% methanol extracts of the unfermented *Toxicodendron vernicifluum* stem bark (UTVSB) (**A**) and 10-day fermented TVSB (**B**).

**Figure 6 molecules-24-00683-f006:**
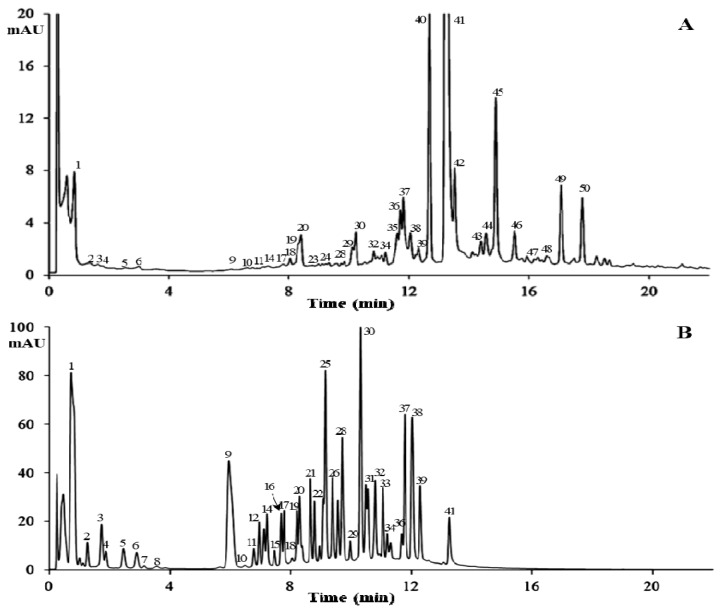
Total ion current (TIC) chromatograms of 10-day fermented TVSB (**A**) and PGG hydrolysate (**B**) by a crude enzyme preparation isolated from fermented TVSB.

**Figure 7 molecules-24-00683-f007:**
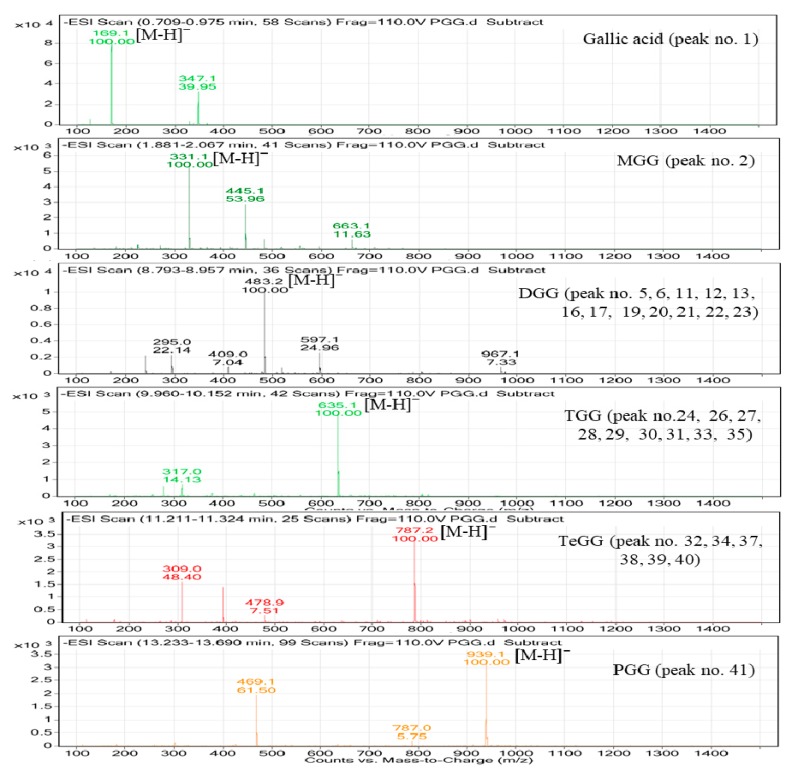
LC-MS spectra of PGG and its hydrolysates identified in fermented TVSB.

**Figure 8 molecules-24-00683-f008:**
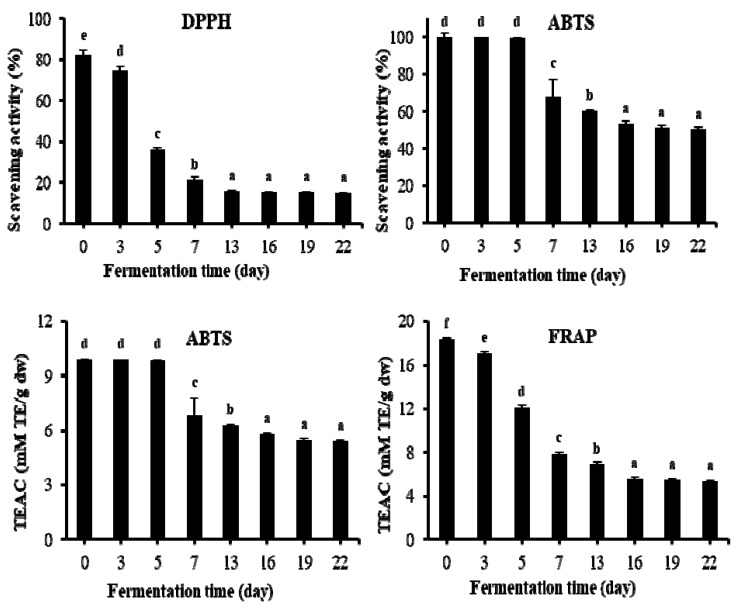
Changes of antioxidant activities during fermentation of TVSB by *F. fraxinea*. Different letters indicate values that are significantly different by Duncan’s multiple range test at 5% level. TEAC, Trolox equivalent antioxidant capacity. DPPH, 2,2-diphenyl-1-picrylhydrazyl; ABTS, 2,2′-azino-bis-3-ethylbenzothiazoline-6-sulfonic acid; FRAP, ferric reducing antioxidant power. Error bars are the standard deviations of triplicate measurements.

**Table 1 molecules-24-00683-t001:** Compounds detected in fermented TVSB and PGG hydrolysate by crude enzyme preparation isolated from fermented TVSB.

Peak No.	tR (min)	UV (λ_max_, nm)	[M − H]^−^ (*m/z*)	Other Fragment (*m/z*)	Identification	Detection	
						FTVSB	PGGH
1	0.714	272	169.0	331.1	Gallic acid	O	X
2	1.264	276	331.1	445.1, 271.0	MGG	O	O
3	1.699	276	483.1	597.0, 224.9	DGG	O	O
4	1.890	276	331.1	445.1, 270.9	MGG (isomer)	O	O
5	2.371	276	483.1	597.0	DGG (isomer)	O	O
6	2.781	273	483.1	181.0, 597.0	DGG (isomer)	O	O
7	3.141		255.1	123.0	Unidentified	X	O
8	3.500		301.0	415.0, 197.8	Unidentified	O	O
9	5.848	276	183.1	375.1	Methyl gallate	O	O
10	6.464		283.0	396.8	Unidentified	O	O
11	6.744	276	483.1	241.0, 596.9	DGG (isomer)	O	O
12	6.950	275	483.1	597.1, 241.1	DGG (isomer)	X	O
13	7.118	273	483.1	597.0, 241.0	DGG (isomer)	O	O
14	7.234	276	225.1	483.1, 339.0	Unidentified	O	O
15	7.449		283.1	397.0, 575.0	Unidentified	X	O
16	7.692	275	483.1	241.1, 597.0	DGG (isomer)	X	O
17	7.799	275	483.1	597.0, 241.1	DGG (isomer)	O	O
18	7.991		283.1	397.1	Unidentified	O	O
19	8.215	277	483.1	241.0, 597.0	DGG (isomer)	O	O
20	8.313	277	483.1	597.0, 241.1	DGG (isomer)	O	O
21	8.658	216, 273	483.1	597.1, 241.0	DGG (isomer)	X	O
22	8.803	216, 273	483.2	597.0, 295.0	DGG (isomer)	X	O
23	8.962	275	483.2	597.1, 241.0	DGG (isomer)	O	O
24	9.097	276	635.1	749.1, 317.1	TGG	O	O
25	9.111	216, 279	431.2	499.1, 563.1	Unidentified	X	O
26	9.386	216, 276	635.1	749.1, 317.2	TGG (isomer)	O	O
27	9.578	217, 276	635.1	317.1	TGG (isomer)	O	O
28	9.708	216, 276	635.1	749.1, 317.1	TGG (isomer)	O	O
29	9.979	216, 276	635.1	317.0, 169.0	TGG (isomer)	O	O
30	10.315	217, 278	635.1	317.1, 169.0	TGG (isomer)	O	O
31	10.502	217, 278	635.1	317.2	TGG (isomer)	O	O
32	10.777	217, 278	787.1	635.1, 393.1	TeGG	O	O
33	11.057	217, 278	635.1	317.1, 749.1	TGG (isomer)	O	O
34	11.225	218, 277	787.1	393.2, 309.0	TeGG (isomer)	O	O
35	11.328	217, 278	635.1	317.2, 749.0	TGG (isomer)	X	O
36	11.688	217, 238	301.0	610.9	Ellagic acid	O	O
37	11.790	217, 278	787.1	393.2, 301.1	TeGG (isomer)	O	O
38	12.005	217, 279	787.1	393.2	TeGG (isomer)	O	O
39	12.280	217, 279	787.1	393.1	TeGG (isomer)	O	O
40	13.195	217, 279	787.1	393.1	TeGG (isomer)	O	X
41	13.256	217, 279	939.1	469.1	PGG	O	O
42	13.503	316, 360	285.1	113.0	Fisetin	O	X
43	14.344		255.1	432.9	Unidentified	O	X
44	14.549		401.0	287.2, 723.4	Unidentified	O	X
45	14.806		209.1	539.1	Unidentified	O	X
46	15.441	265, 264	301.1	415.1, 603.0	Quercetin	O	X
47	16.430		423.1	271.1, 536.9	Unidentified	O	X
48	16.995	261, 379	271.1	551.1, 385.1	Butein	O	X
49	21.014		417.1	531.0	Unidentified	O	X
50	23.105		653.0	539.0, 518.7	Unidentified	O	X

DGG: digalloylglucose; FTVSB, 10-days fermented TVSB; MGG: monogalloylglucose; PGGH, PGG hydrolysate; O, detected; TeGG: tetragalloylglucose; TGG: trigalloylglucose; X, not detected.

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
