# Peer review of "Changes of Phytochemical Components (Urushiols, Polyphenols, Gallotannins) and Antioxidant Capacity during Fomitella fraxinea–Mediated Fermentation of Toxicodendron vernicifluum Bark"

_molecules, 2019, doi:10.3390/molecules24040683_

Round 1

Reviewer 1 Report

The title must be reformulated, it is much complex.

The introduction reports several previous studies what does this adds to those already published? I understand the focus on the dynamic changes but authors must strain the worth of the present manuscript.

Please explain better the implications of this:  "It has been previously reported that basidiomycete polyphenol oxidases can effectively oxidize flavonoids such as quercetin, fisetin, rutin, luteolin, myricetin and kaempferol, as well as phenolic acids"

In discussion the authors provide a small theoretical introduction for each parameter, in my opinion it would be better to replace this by a more strict comparison to results obtained by other authors in the same matrix or in similar ones.

Where the plants tested for the presence of pesticides? Or microbial contamination?

Add a paragraph regarding the future implications and uses of the reported discoveries at the end of the conclusion; the available information is not enough.

Author Response

Dear Editor and Reviewers, 

We highly appreciate the detailed and valuable comments of the reviewers on our manuscript of Manuscript ID: molecules-422172, Dynamic Changes of Urushiols, Polyphenols, Hydrolysable Tannins and Antioxidant Capacity during Fomitella fraxinea–mediated Fermentation of Toxicodendron vernicifluum Bark. We have revised our manuscript in response to their comments and suggestions. On behalf of my co-authors, I would like to clarify some of the points raised by the reviewers below. We hope the reviewers and editors will be satisfied with our responses to the ‘comments and suggestions’ and the revisions for the original manuscript.

With best regards!

Yours Sincerely,

M. K. Kim

Department of Food Science and Technology, Chonbuk National University, Jeonju,

Jeollabuk-do 54896, Republic of Korea

E-mail: kmyuko@jbnu.ac.kr

   Author’s responses for reviewer's comments and suggestions

Manuscript No: molecules-422172

Title: Dynamic Changes of Urushiols, Polyphenols, Hydrolysable Tannins and Antioxidant Capacity during Fomitella fraxinea–mediated Fermentation of Toxicodendron vernicifluum Bark

Authors: Myung-Kon Kim et al.

Correspondence: kmyuko@jbnu.ac.kr (M. K. K.); Tel.: +82-63-270-2551 (M. K. K)\

Reviewer’s comments 1. The title must be reformulated, it is much complex.

Author’s response: According to reviewer’s comments, the title was reformulated as follows.

Changes of phytochemical components (Urushiols, Polyphenols, Gallotannins) and Antioxidant Capacity during Fomitella fraxinea–mediated Fermentation of Toxicodendron vernicifluum Bark’

Reviewer’s comments 2. The introduction reports several previous studies what does this adds to those already published? I understand the focus on the dynamic changes but authors must strain the worth of the present manuscript.

Author’s response: We included more previous results as a background to emphasize the research problem. In agreement with the reviewer, we have added the following paragraph in conclusion section to explain significance of our results from the research problem point of view. (Lines 461-467 in revised manuscript)

Although TVSB extracts are fermented for 2028 days using F. fraxinea to be marketed as functional food raw materials in Korea, the present study suggests that the optimal fermentation period to remove or reduce toxic urushiols is between 13−16 days with a minimized reduction of useful constituents such as flavonoids and gallotannins.’ The findings in the present work also provide clear evidences for the need of additional studies on enzymatic characteristics of PGGhydrolyzing enzymes secreted from F. fraxinea during fermentation of TVSB.’

Reviewer’s comments 3. Please explain better the implications of this: "It has been previously reported that basidiomycete polyphenol oxidases can effectively oxidize flavonoids such as quercetin, fisetin, rutin, luteolin, myricetin and kaempferol, as well as phenolic acids"

Author’s response: As per the comment of the reviewer, we have elaborated it as follows. (lines 133-137 in revised manuscript)

It has been previously reported that basidiomycete polyphenol oxidases can effectively utilize flavonoids such as quercetin, fisetin, rutin, luteolin, myricetin and kaempferol, as well as phenolic acids as substrates [28, 31, 33, 34]. Therefore, the reduction of flavonoids observed during frementation of the TVSB extract is suggested to be mediated by polyphenol oxidases secreted from F. fraxinea.’

Reviewer’s comments 4. In discussion the authors provide a small theoretical introduction for each parameter, in my opinion it would be better to replace this by a more strict comparison to results obtained by other authors in the same matrix or in similar ones.

Author’s response: As the reviewer pointed out, results should be discussed with previous data. However, there is no previous data available on all parameters we analyzed except for Changes of urushiols which we have modified the discussion as follows. (Lines 91-94 in revised manuscript)

“Choi et al. [19] who attempted similar biological method for the reduction of urushiols reported that content of C15:3 decreased rapidly in the first 5 days of fermentation followed by a gradual decrease reaching the lowest levels after 15 days of fermentation.”

Reviewer’s comments 5. Where the plants tested for the presence of pesticides? or microbial contamination?

Author’s response: ‘The Toxicodendron vernicifluum trees used in this study were mainly cultivated in an uncontaminated farm and pesticides are not used during cultivation of T. vernicifluum tree in Korea and the TVSB samples were sterilized (121 ℃, 30 min) prior to inoculation with F. fraxinea cultures. Therefore, the possibility of contamination with pesticides and microbes is believed to be negligible and no test was conducted for pesticides.’

Reviewer’s comments 6. Add a paragraph regarding the future implications and uses of the reported discoveries at the end of the conclusion; the available information is not enough.

Author’s response: In agreement with the reviewer the following paragraph is supplemented in line 461 of revised manuscript.

‘Although TVSB extracts are being fermented for 2028 days using F. fraxinea to be marketed as functional food raw materials in Korea, the present study suggests that the optimal fermentation period to remove or reduce toxic urushiols was between 13−16 days with a minimized reduction of useful constituents such as flavonoids and gallotannins.’ The findings in the present work also provide clear evidences for the need of additional studies on enzymatic characteristics of PGGhydrolyzing enzymes secreted from F. fraxinea during fermentation of TVSB.’

Reviewer 2 Report

The manuscript describes presumed fungal (Fomitella fraxinea) fermentation of T. vernicifluum bark resulting in decreasing levels of phenolic compounds over time.  The analytical chemistry results are presented in a very straight forward manner.  The primary conclusions of fungal-specific alternation in phytochemical levels is not demonstrated (at least not clearly) because of the absence of a matched control in which the sterilize bark sample is sampled over the same time frame as the fungal-treated cultures.  All references to Unfermented TVSB (UTVSB) appear to be a single time point (at time equal to zero/beginning).  Thus, there is no control(s) to rule out the possibility that either heat stable Toxicodendron enzymes or autocatalytic phytochemical degradation is responsible for some or all of the observed decreases in phenolic compounds.  Without this control, the assertion of fungal-specific changes in the phytochemicals is not unambiguously validated. 

The Materials and Methods section is unclear about several aspects that directly impact fundamental assumptions of ANOVA statistical analyses.  First of all, were replicate fermentation cultures initiated at the same time/day, or where they independently established on completely different days/weeks/months?  If they were all set up on the day (I am guessing from the error bars, that was the case) then there were NOT INDEPENDENT replications of the experiments.  The error bars on graphs are not classified as standard deviation or standard error.  Which is it?  Running a one-way ANOVA on time series data is not valid because of the inherent violation of the foundational assumption of Independent sampling in an ANOVA model.  Thus, the experiments violated the one of the foundational assumptions of ANOVA.  This can be corrected by using a nested ANOVA linear regression model, or by redesigning and repeating the experiments to avoid resampling in each culture.

Author Response

Dear Editor and Reviewers, 

We highly appreciate the detailed and valuable comments of the reviewers on our manuscript of Manuscript ID: molecules-422172, Dynamic Changes of Urushiols, Polyphenols, Hydrolysable Tannins and Antioxidant Capacity during Fomitella fraxinea–mediated Fermentation of Toxicodendron vernicifluum Bark. We have revised our manuscript in response to their comments and suggestions. On behalf of my co-authors, I would like to clarify some of the points raised by the reviewers below. We hope the reviewers and editors will be satisfied with our responses to the ‘comments and suggestions’ and the revisions for the original manuscript.

With best regards!

Yours Sincerely,

M. K. Kim

Department of Food Science and Technology, Chonbuk National University, Jeonju,

Jeollabuk-do 54896, Republic of Korea

E-mail: kmyuko@jbnu.ac.kr

Author’s responses for reviewer's comments and suggestions

Manuscript No: molecules-422172

Title: Dynamic Changes of Urushiols, Polyphenols, Hydrolysable Tannins and Antioxidant Capacity during Fomitella fraxinea–mediated Fermentation of Toxicodendron vernicifluum Bark

Authors: Myung-Kon Kim et al.

Correspondence: kmyuko@jbnu.ac.kr (M. K. K.); Tel.: +82-63-270-2551 (M. K. K)

Reviewer’s comments 1. The manuscript describes presumed fungal (Fomitella fraxinea) fermentation of T. vernicifluum bark resulting in decreasing levels of phenolic compounds over time. The analytical chemistry results are presented in a very straight forward manner. The primary conclusions of fungal-specific alternation in phytochemical levels is not demonstrated (at least not clearly) because of the absence of a matched control in which the sterilize bark sample is sampled over the same time frame as the fungal-treated cultures.

Author’s response: We agree with the reviewer that primary conclusions should be demonstrated based on comparisons with matched controls. We are submitting phytochemical levels of the control at days 0 and 22 as supplementary data (Figure S1-S4). We are also submitting TLC images (Supplementary data S1 and S2) of all controls and test samples to strengthen our conclusions. It can be seen from the results that the phytochemical profiles of matched control showed no variation between day 0 and day 22. For this reason, we decided that it is unnecessary to put this data in the main manuscript. We would like to describe that the label ‘control’ in figure 1 A indicates the data for 0 day test sample as well as 0 day control sample as we used the same sample before inoculation for both. We have corrected the label to ‘0 day (Figure 1A).

Reviewer’s comments 2. All references to unfermented TVSB (UTVSB) appear to be a single time point (at time equal to zero/beginning). Thus, there is no control(s) to rule out the possibility that either heat stable Toxicodendron enzymes or autocatalytic phytochemical degradation is responsible for some or all of the observed decreases in phenolic compounds. Without this control, the assertion of fungal-specific changes in the phytochemicals is not unambiguously validated.

Author’s response: We have actually conducted all experiments for the UTVSB and the data for the matched control is included in our new supplementary data (Figure S1-S4). In the TLC images in the supplementary data, it can be clearly observed that the bands of the compounds show no variation in the matched controls (S1 B and S2 B) while there is a steady decrease of band intensity in the fermented samples (S1 A and S2 B) with fermentation time. Additionally, all samples (matched control and test samples) were sterilized (121 , 30 min) prior to inoculation with F. fraxinea culture. Therefore, degradation of phytochemicals by enzymes present in T. vernicifluum bark or autocatalytic phytochemical degradation is very unlikely.

Reviewer’s comments 3. The Materials and Methods section is unclear about several aspects that directly impact fundamental assumptions of ANOVA statistical analyses. First of all, were replicate fermentation cultures initiated at the same time/day, or where they independently established on completely different days/weeks/months? If they were all set up on the day (I am guessing from the error bars, that was the case) then there were NOT INDEPENDENT replications of the experiments. The error bars on graphs are not classified as standard deviation or standard error. Which is it?   Running a one-way ANOVA on time series data is not valid because of the inherent violation of the foundational assumption of Independent sampling in an ANOVA model. Thus, the experiments violated the one of the foundational assumptions of ANOVA. This can be corrected by using a nested ANOVA linear regression model, or by redesigning and repeating the experiments to avoid resampling in each culture.

Author’s response:  A description indicating that the error bars are standard deviations is added in the figure legends of all graphs in the revised manuscript.  Regarding the statistical validity and sampling, air-dried and powdered TVSB samples were distributed into separate flasks for each sample type (0, 3, 5, 7, 10, 13, 16, 19 & 22 days fermented/unfermented (matched control) with triplicate. 0 day control sample was immediately stored in -20C while the remaining samples (fermented and matched control samples) were incubated for each required time frame. In order to investigate the reproducibility of the results, similar fermentation experiments were performed repeatedly in different months. The reduction ratio of the constituents (urushiols, total phenol, total flavonoids and individual flavonoids) and antioxidant properties showed similar patterns although standard deviation was somewhat different. This shows that the experimental sampling was actually independent and we believe it is scientifically valid to run one way independent samples ANOVA based on the independent variable ‘fermentation time’. Our response could be somewhat different from the reviewer's point of view, but we will appreciate your understanding on our response. To clarify the method more we have added the following content in lines 275-282 of the new manuscript.

The moisture content of each powdered sample (100g) was then adjusted to approximately 65% with tap water. Each flask was sterilized at 121 °C for 30 min. 0 day (control) sample was freeze-dried immediately and stored at -20C until further analysis. The remaining samples were inoculated with 5 mL of F. fraxinea liquid culture except the unfermented (matched controls) and incubated at room temperature (2526 °C) with gentle shaking (120 rpm for each required time frame before further analysis.” Each sample was then freeze-dried at the end of each time frame and stored until use.

Reviewer 3 Report

The work “Dynamic Changes of Urushiols, Polyphenols, Hydrolysable Tannins and Antioxidant Capacity during Fomitella fraxinea – mediated Fermentation of Toxicodendron vernicifluum Bark is topical, interesting and in general very well written. Article require some corrections before publishing in Molecules.

Line 61. Authors should standardize the type.

Line 69. Authors should standardize the type.

Line 107. Authors should remove attenuation.

Line 150. Authors should standardize the type.

Figure 5. The letter ‘a’ in the word “and” should not be bold.

Line 192. Authors should standardize the type.

Line 199. Authors should standardize the type.

Table 2. Authors should standardize the type in the description.

Table 2. Authors should correct the word “fermentation” and get the description and results in columns right.

Line 256. Authors should write some information about characteristics of the strain.

Line 265. What kind of shaker did you used? Please, add the information.

Line 267. Authors should correct the unit of Celsius degree.

Line 315. Consider write the results in the range without space: 12.5-750, not 12.5 – 750. Please, check the whole manuscript.

Line 335. What kind of centrifuge did you used? Please, add the information.

Line 407. What kind of shaker did you used? Please, add the information.

Line 411. Authors should standardize the type – check the entire manuscript.

Line 418. Consider using figures in the whole manuscript: 100 µL, not One hundred.

Line 428. Separate the figure from the unit.

Line 432. Full stop is required.

Author Response

Dear Editor and Reviewers, 

We highly appreciate the detailed and valuable comments of the reviewers on our manuscript of Manuscript ID: molecules-422172, Dynamic Changes of Urushiols, Polyphenols, Hydrolysable Tannins and Antioxidant Capacity during Fomitella fraxinea–mediated Fermentation of Toxicodendron vernicifluum Bark. We have revised our manuscript in response to their comments and suggestions. On behalf of my co-authors, I would like to clarify some of the points raised by the reviewers below. We hope the reviewers and editors will be satisfied with our responses to the ‘comments and suggestions’ and the revisions for the original manuscript.

With best regards!

Yours Sincerely,

M. K. Kim

Department of Food Science and Technology, Chonbuk National University, Jeonju,

Jeollabuk-do 54896, Republic of Korea

E-mail: kmyuko@jbnu.ac.kr

Author’s responses for reviewer's comments and suggestions

Manuscript No: molecules-422172

Title: Dynamic Changes of Urushiols, Polyphenols, Hydrolysable Tannins and Antioxidant Capacity during Fomitella fraxinea–mediated Fermentation of Toxicodendron vernicifluum Bark

Authors: Myung-Kon Kim et al.

Correspondence: kmyuko@jbnu.ac.kr (M. K. K.); Tel.: +82-63-270-2551 (M. K. K)

Reviewer’s comments 1.  Line 61. Authors should standardize the type.

Author’s response: Line 62 in the revised manuscript is corrected as suggested by reviewer.

Reviewer’s comments 2. Line 69. Authors should standardize the type.

Author’s response: Line 69 in the revised manuscript is corrected as suggested by reviewer.

Reviewer’s comments 3. Line 107. Authors should remove attenuation.

Author’s response: Line 111 in the revised manuscript is corrected as suggested by reviewer.

Reviewer’s comments 4. Line 150. Authors should standardize the type.

Author’s response: Lines 160-161 in the revised manuscript are corrected as suggested by reviewer.

Reviewer’s comments 5. Figure 5. The letter ‘a’ in the word “and” should not be bold.

Author’s response: The letter ‘a’ in caption of Figure 5 is corrected.

Reviewer’s comments 6. Line 192. Authors should standardize the type.

Author’s response: : Line 199 in the revised manuscript is corrected as suggested by reviewer.

Reviewer’s comments 7. Line 199. Authors should standardize the type.

Author’s response: : Line 206 in the revised manuscript is corrected as suggested by reviewer.

Reviewer’s comments 8. Table 2. Authors should standardize the type in the description.

Author’s response: Line 242 in the revised manuscript (Table 2) is corrected as suggested by reviewer.

Reviewer’s comments 9. Table 2. Authors should correct the word “fermentation” and get the description and results in columns right.

Author’s response: Line 242 in the revised manuscript (Table 2) is corrected as suggested by reviewer.

Reviewer’s comments 10. Line 256. Authors should write some information about characteristics of the strain.

Author’s response: If this suggestion is about the fungi strain, some background information is included in the introduction section and the specific strain is clearly indicated in the methodology part. Regarding the Toxicodendron vernicifluum strain, no strain is characterized or established in Korea. We simply collected the sample from a random local farm.

Reviewer’s comments 10. Line 265. What kind of shaker did you used? Please, add the information.

Author’s response:  Description of shaker’s model is added in line 275 of the revised manuscript as follow

 “Using orbital shaker (model SK-600, Jeiotech Co., LTD. Daejon, Korea).

Reviewer’s comments 11. Line 267. Authors should correct the unit of Celsius degree.

Author’s response: The unit of Celsius degree is corrected.

Reviewer’s comments 12. Line 315. Consider write the results in the range without space: 12.5-750, not 12.5 – 750. Please, check the whole manuscript.

Author’s response: Space ranges are corrected as follow.

‘(12.5 750 μg/mL)’ → ‘(12.5750 μg/mL)’

Reviewer’s comments 13. Line 335. What kind of centrifuge did you used? Please, add the information.

Author’s response: The specification of centrifuge is added in line 346 as follow.

‘(model VS-550, Vision Scientific Co., LTD, Daejon, Korea)’

Reviewer’s comments 14. Line 407. What kind of shaker did you used? Please, add the information.

Author’s response: ‘Shaken’ is corrected to ‘incubated’ in line 419..

Reviewer’s comments 15. Line 411. Authors should standardize the type – check the entire manuscript.

Author’s response: The font in line 421 is corrected..

Reviewer’s comments 16. Line 418. Consider using figures in the whole manuscript: 100 µL, not One hundred.

Author’s response: one hundred uL in line 429 →100 uL.

Reviewer’s comments 17. Line 428. Separate the figure from the unit.

Author’s response: 20mM → 20 mM.

Reviewer’s comments 18. Line 432. Full stop is required.

Author’s response: Full stop is added in line 432.

Reviewer 4 Report

The manuscript describes the dynamic changes of total phenol content, individual bioactive compounds, PGG and allergic compounds of the stem bark of Toxicodendron vernicifluum (TVSB) after fermented by Formitella fraxinea to reduce allergic compounds. The authors provided sufficient evidence to support their claims, and the manuscript was well prepared. There is one issue should be revised before publishing. In page 6, line 160, the results of the total ion current chromatograph were summarized in figure 6. The authors can change “figure 6A” just to “figure 6” since the authors did not indicate “figure 6B” in the article.  

Author Response

Dear Editor and Reviewers, 

We highly appreciate the detailed and valuable comments of the reviewers on our manuscript of Manuscript ID: molecules-422172, Dynamic Changes of Urushiols, Polyphenols, Hydrolysable Tannins and Antioxidant Capacity during Fomitella fraxinea–mediated Fermentation of Toxicodendron vernicifluum Bark. We have revised our manuscript in response to their comments and suggestions. On behalf of my co-authors, I would like to clarify some of the points raised by the reviewers below. We hope the reviewers and the editors will be satisfied with our responses to the ‘comments and suggestions’ and the revisions for the original manuscript.

With best regards!

Yours Sincerely,

M. K. Kim

Department of Food Science and Technology, Chonbuk National University, Jeonju,

Jeollabuk-do 54896, Republic of Korea

E-mail: kmyuko@jbnu.ac.kr

Author’s responses for reviewer's comments and suggestions

Manuscript No: molecules-422172

Title: Dynamic Changes of Urushiols, Polyphenols, Hydrolysable Tannins and Antioxidant Capacity during Fomitella fraxinea–mediated Fermentation of Toxicodendron vernicifluum Bark

Authors: Myung-Kon Kim et al.

Correspondence: kmyuko@jbnu.ac.kr (M. K. K.); Tel.: +82-63-270-2551 (M. K. K)

Reviewer’s comments 1. The manuscript describes the dynamic changes of total phenol content, individual bioactive compounds, PGG and allergic compounds of the stem bark of Toxicodendron vernicifluum (TVSB) after fermented by Formitella fraxinea to reduce allergic compounds. The authors provided sufficient evidence to support their claims, and the manuscript was well prepared. There is one issue should be revised before publishing. In page 6, line 160, the results of the total ion current chromatograph were summarized in figure 6. The authors can change “figure 6A” just to “figure 6” since the authors did not indicate “figure 6B” in the article.

Author’s response: Figure 6B’ is corrected to ‘Figure 6.’

Round 2

Reviewer 1 Report

This sentence should be rewritten, it is redundant and not clear: "The stem bark of Toxicodendron vernicifluum (TVSB) has been traditionally used as traditional herbal medicine and health foods in Korea."

" However, its application in food and 14 pharmaceutical industry has been restricted due to its potential to cause allergy." The problem is just clergy?? What about the eventual contamination by pesticides and the lack of studies regarding its composition?

In the end of introduction add a paragraph regarding the importance and novelty of the study.

"This mechanism leads to detoxification by reducing active site of urushiols" - please explain better.

"Because of the fact that phytochemicals exhibit antioxidant activities by different mechanisms depending on their chemical structures, overall antioxidant property of plant samples cannot be accurately evaluated by any single method" - provide more information.

Overall, the resolution of the figures and graphs must be improved.

Plants were harvested in 2015 - why does it took so long to submit the manuscript? Also, "The fresh TVSB was air air‐dried for 24 h and kept at room temperature until usage." - how much time was the product left at room temperature?

Author Response

Dear Editor and Reviewers, 

We highly appreciate the second comments of the reviewers on our manuscript of Manuscript ID: molecules-422172, Changes of     Phytochemical Components (Urushiols, Polyphenols, Gallotannins) and Antioxidant Capacity during Fomitella fraxineamediated Fermentation of Toxicodendron vernicifluum Bark. We have revised our manuscript one more time in response to the second round comments and suggestions. We hope the reviewers and the editors will be satisfied with our responses to the ‘comments and suggestions’ and the revisions for the original manuscript.

With best regards!

Yours Sincerely,

M. K. Kim

Department of Food Science and Technology, Chonbuk National University, Jeonju,

Jeollabuk-do 54896, Republic of Korea

E-mail: kmyuko@jbnu.ac.kr

Author’s responses for reviewer's comments and suggestions

Manuscript No: molecules-422172

Title: Changes of Phytochemical Components (Urushiols, Polyphenols, Gallotannins) and Antioxidant Capacity during Fomitella fraxineamediated Fermentation of Toxicodendron vernicifluum Bark

Authors: Myung-Kon Kim et al.

Correspondence: kmyuko@jbnu.ac.kr (M. K. K.); Tel.: +82-63-270-2551 (M. K. K)

NB: We would like to inform you the second round revisions are highlighted in green to differentiate them from the first round revisions which are highlighted in yellow.  Line numbers are according to the revised version of manuscript.

Reviewer’s comments 1; This sentence should be rewritten, it is redundant and not clear: "The stem bark of Toxicodendron vernicifluum (TVSB) has been traditionally used as traditional herbal medicine and health foods in Korea."

Author’s response: We have rewritten the sentence in line 13-14 as follows as noted by the reviewer.

The stem bark of Toxicodendron vernicifluum (TVSB) has been widely used as traditional herbal medicine and food ingredients in Korea.

Reviewer’s comments 2; " However, its application in food and 14 pharmaceutical industry has been restricted due to its potential to cause allergy." The problem is just clergy?? What about the eventual contamination by pesticides and the lack of studies regarding its composition?

Author’s response: The following sentence is added in line 15-16 to describe more research problem as suggested by the reviewer.

“. Moreover, there is limited data available on the qualitative and quantitative changes in composition of its phytochemicals during fermentation.”

Reviewer’s comments 3; In the end of introduction add a paragraph regarding the importance and novelty of the study.

Author’s response: As suggested by reviewer, the following paragraph is added from line 65-70.

 “In this study we demonstrated optimization of an effective detoxification of urushiol congeners in TVSB through monitoring fermentation time while simultaneously intending to achieve a minimized loss of other useful phenolic compounds. The optimal fermentation period to achieve the desired detoxification effect was found to be 13 – 16 days. This finding provides valuable information to reconsider the common practice of fermenting TVSB extracts with F. fraxinea for 20 – 28 days prior to their applications in Korea. 

Reviewer’s comments 4; "This mechanism leads to detoxification by reducing active site of urushiols" - please explain better.

Author’s response: In agreement with the reviewer, this concept is elaborated as follows from line 112-117.

 However, laccase can oxidize urushiols and result in the formation of semiquinone radical under aerobic conditions. This is a typically unstable product and it undergoes subsequent changes either through an attack on the urushiol nucleus forming biphenyl compounds or disproportionation reaction to give urushiol quinone [29]. It may also undergo polymerization reactions which results in the formation of a conjugated insoluble product. All these mechanisms lead to reduction of the active site of urushiols thereby resulting in its detoxification [29,30].

Reviewer’s comments 5;"Because of the fact that phytochemicals exhibit antioxidant activities by different mechanisms depending on their chemical structures, overall antioxidant property of plant samples cannot be accurately evaluated by any single method" - provide more information.

Author’s response: As suggested by the reviewer, the content at the beginning of section 2.5 is revised from 218-227 to provide more information as follows;

Due to the differences in the theoretical bases of different antioxidant measurements, single antioxidant property model can hardly reflect the antioxidant capacity of samples [43]. For this reason, three model systems, i.e. DPPH radical scavenging activity, ABTS radical scavenging activity and FRAP were used to evaluate the antioxidant properties of fermented TVSB. DPPH and ABTS rely on the reaction of radicals and cation radicals, respectively and the FRAP method relies on the reduction by the antioxidant components of complex ferric ion – TPTZ (2,4,6-tri(2-pyridyl)-s- triazine). A study by Gorinstein et al. [44] observed high correlation of antioxidant capacities measured by ABTS, DPPH and FRAP assays in some fruits while Pellegrini et al. [45] reported a weak correlation on antioxidant capacity measured by FRAP and ABTS assays in some vegetables and beverages. The results of changes in antioxidant capacities are presented in Figure 8.

Reviewer’s comments 6; Overall, the resolution of the figures and graphs must be improved.

Author’s response: Figures 5 and 6 were replaced with better resolution ones as suggested by reviewer.

Reviewer’s comments 7; Plants were harvested in 2015 - why does it took so long to submit the manuscript? Also, "The fresh TVSB was air airdried for 24 h and kept at room temperature until usage." - how much time was the product left at room temperature?

Author’s response: The submission was delayed because of tight schedules and the long time we took searching for an appropriate journal with a fitting scope. To clarify the indicated method, the sentence in line 268-269 is revised as follows.

“The fresh TVSB was air‐dried (60 °C) for 24 h and kept at room temperature (20 °C) until used for further experiments within 1 week.

Reviewer 2 Report

The supplemental figures do much to address concerns about fungal specific reduction in T. vernicifluum polyphenol and alkylphenol levels.  The response to the editor mentions there was a separate fermentation reaction for each time point, which is an appropriate means of eliminating sample time-(in)dependence concerns.  However, this is not clearly stated in the revised methods section.  Moreover, it seems that the proper negative control was performed only once.

The revised manuscript still does not state whether the three fermentation replications were set up on the same day or on different dates.  This needs to be explicitly stated in the methods section. 

Author Response

Dear Editor and Reviewers, 

We highly appreciate the second comments of the reviewers on our manuscript of Manuscript ID: molecules-422172, Changes of     Phytochemical Components (Urushiols, Polyphenols, Gallotannins) and Antioxidant Capacity during Fomitella fraxineamediated Fermentation of Toxicodendron vernicifluum Bark. We have revised our manuscript one more time in response to the second round comments and suggestions. We hope the reviewers and the editors will be satisfied with our responses to the ‘comments and suggestions’ and the revisions for the original manuscript.

With best regards!

Yours Sincerely,

M. K. Kim

Department of Food Science and Technology, Chonbuk National University, Jeonju,

Jeollabuk-do 54896, Republic of Korea

E-mail: kmyuko@jbnu.ac.kr

Author’s responses for reviewer's comments and suggestions

Manuscript No: molecules-422172

Title: Changes of Phytochemical Components (Urushiols, Polyphenols, Gallotannins) and Antioxidant Capacity during Fomitella fraxineamediated Fermentation of Toxicodendron vernicifluum Bark

Authors: Myung-Kon Kim et al.

Correspondence: kmyuko@jbnu.ac.kr (M. K. K.); Tel.: +82-63-270-2551 (M. K. K)

NB: We would like to inform you the second round revisions are highlighted in green to differentiate them from the first round revisions which are highlighted in yellow.  Line numbers are according to the revised version of manuscript.

Reviewer’s comment 1;The supplemental figures do much to address concerns about fungal specific reduction in T. vernicifluum polyphenol and alkylphenol levels.  The response to the editor mentions there was a separate fermentation reaction for each time point, which is an appropriate means of eliminating sample time-(in)dependence concerns.  However, this is not clearly stated in the revised methods section.  Moreover, it seems that the proper negative control was performed only once.

Author’s response: As we have mentioned before, we have incubated the control for the same duration as the fermented sample. We also did sampling of the control for all time points as matched controls. However, we realized that the TLC results have no difference for all time points of the matched controls (0 -22 days). Due to this, we performed HPLC analysis only for two of the matched controls (0 & 22 days incubation) with no difference observed. As pointed out by the reviewer, we have added section 3.5.2 in methods and included the following information in line 87-90 of the revised manuscript as follows.

“The control was incubated for the same time intervals as the fermented sample. However, HPLC analysis of the controls was carried out only for 0 and 22 day time points because thin-layer chromatography (TLC) results (Supplementary data figure S1–S4) showed no difference among all matched controls”

Reviewer’s comment 2; The revised manuscript still does not state whether the three fermentation replications were set up on the same day or on different dates. This needs to be explicitly stated in the methods section.

Author’s response: The following sentence is added in line 297-298 according to the reviewer’s comment.

All fermentations were set up on the same day and each sample for the specific time point was retrieved with its matched control at the same time.